# Unraveling Pneumomediastinum in COVID-19 Patients: Insights from a High-Volume-Center Case–Control Study

**DOI:** 10.3390/diseases12100242

**Published:** 2024-10-03

**Authors:** Khrystyna Kuzmych, Marcello Covino, Mattia Paratore, Annalisa Campanella, Ludovico Abenavoli, Giuseppe Calabrese, Antonio Giulio Napolitano, Carolina Sassorossi, Stefano Margaritora, Filippo Lococo

**Affiliations:** 1Thoracic Surgery Unit, Università Cattolica del Sacro Cuore, Largo F. Vito 1, 00168 Rome, Italy; kkristina.kuz@gmail.com (K.K.); annalisa.campanella@guest.policlinicogemelli.it (A.C.); giuseppe93calabrese@virgilio.it (G.C.); antoniogiulionapolitano@gmail.com (A.G.N.); sassorossi.caro@gmail.com (C.S.); stefano.margaritora@policlinicogemelli.it (S.M.); 2Emergency Department, Fondazione Policlinico Universitario Gemelli IRCCS, 00168 Roma, Italy; marcello.covino@policlinicogemelli.it; 3Diagnostic and Interventional Ultrasound Unit, CEMAD Digestive Disease Center, Internal Medicine and Gastroenterology, Catholic University of the Sacred Heart, Fondazione Policlinico Universitario Gemelli IRCCS, 00168 Rome, Italy; mattia.paratore@guest.policlinicogemelli.it; 4Department of Health Sciences, University “Magna Græcia”, 88100 Catanzaro, Italy; l.abenavoli@unicz.it; 5Thoracic Surgery Unit, Fondazione Policlinico Universitario Gemelli IRCCS, 00168 Rome, Italy

**Keywords:** pneumomediastinum, COVID-19, respiratory failure

## Abstract

Background: Pneumomediastinum (PNM) is a severe complication in COVID-19 patients, potentially exacerbating morbidity and requiring heightened clinical attention. This study aims to identify risk factors, clinical characteristics, and outcomes associated with PNM in COVID-19 patients hospitalized for respiratory failure in our institution. Methods: Among 4513 patients admitted in our institution and testing positive for COVID-19 infection during the peak of the COVID-19 pandemic in Italy (1 March 2020 to 31 July 2020), we conducted a single-center, retrospective case–control study focusing our analysis on those with severe disease (respiratory failure). The cohort included a total of 65 patients (32 with PNM and 33 without PNM in the same period). Data were retrospectively collected from hospital records, including demographics, comorbidities, smoking history, clinical and laboratory findings, and imaging results. Statistical analyses were performed using Fisher’s exact test and Student’s *t*-test, with significance set at α = 0.05. Results: Patients with PNM were significantly younger (54.9 ± 18.5 vs. 65.4 ± 14.3 years, *p* = 0.0214) and exhibited higher inflammatory markers, particularly white blood cells count (WBC) at admission (11.4 ± 5.4 vs. 6.5 ± 4.1, *p* < 0.0001). Although smoking status, body mass index (BMI), and major comorbidities did not differ significantly between groups, COPD was more prevalent in the PNM group (46.9% vs. 15.1%, *p* = 0.0148). Radiologically, ground-glass opacities (GGOs) and consolidations were more frequent in PNM patients (93.7% vs. 51.5%, *p* = 0.0002; 78.1% vs. 42.2%, *p* = 0.0051, respectively). PNM was associated with longer hospital stays (28.5 ± 14.9 vs. 12.0 ± 7.2 days, *p* < 0.0001) and a higher need for invasive mechanical ventilation (53.1% vs. 30.3%, *p* = 0.0619). However, mortality rates did not differ significantly between groups. Conclusions: PNM in patients with severe COVID-19 infection is associated with younger age, elevated inflammatory markers, and extensive lung involvement, contributing to increased morbidity and prolonged hospitalization. Early detection and tailored management strategies, including optimized respiratory support and aggressive anti-inflammatory therapies, are crucial in mitigating the adverse outcomes associated with PNM. Further research is needed to validate these findings and improve clinical protocols for managing this complication.

## 1. Introduction

The COVID-19 pandemic, caused by the SARS-CoV-2 virus, has presented unprecedented challenges to global healthcare systems. While the majority of COVID-19 cases are mild to moderate, a significant proportion of patients develop severe respiratory complications, leading to acute respiratory distress syndrome (ARDS) and requiring advanced respiratory support. Among these complications, pneumomediastinum (PNM)—the presence of air in the mediastinum—is an increasingly recognized, albeit rare, consequence of severe COVID-19 pneumonia [1]. PNM in COVID-19 patients can arise spontaneously or as a result of barotrauma from mechanical ventilation and is often associated with a poor prognosis due to its association with extensive lung damage [2].

Despite its clinical significance, the pathogenesis and risk factors for PNM in COVID-19 patients remain poorly understood. The existing literature suggests that younger age, severe lung involvement, and underlying pulmonary conditions such as chronic obstructive pulmonary disease (COPD) may predispose individuals to this complication [3]. However, the specific contributions of these factors, along with the role of systemic inflammation and mechanical ventilation, require further investigation to develop effective preventive and therapeutic strategies.

This study aims to elucidate the clinical characteristics, risk factors, and outcomes associated with PNM in patients with severe COVID-19 disease by comparing a cohort of patients who developed PNM with a control group of similar patients with severe COVID-19 disease who did not develop this complication. Through this comparative analysis, we seek to enhance the understanding of PNM’s pathophysiology in the context of COVID-19, ultimately contributing to improved management and patient outcomes.

## 2. Materials and Methods

This is a single-center, retrospective, case–control study. In Figure 1, the consort diagram presents a flow chart of the analyzed population.

In detail, between 1 March 2020 and 31 July 2020, during the peak of the COVID-19 pandemic in Italy, a total of 4513 patients were admitted to our institution with confirmed COVID-19 infection. Among them, we focused our analysis on the 65 patients who presented to the emergency department with severe respiratory failure due to COVID 19 infection and were subsequently admitted to critical care units. The cohort included 32 patients who developed PNM during their hospital stay and 33 who did not.

The diagnosis was confirmed through real-time reverse transcriptase polymerase chain reaction (RT-PCR) on nasopharyngeal swabs. Patients’ information was obtained from the hospital records by a retrospective review. Demographic information, comorbidities, a history of smoking, clinical and laboratory findings [C-reactive protein (CRP), white blood cells (WBCs), D-dimer, for each at admission and the worst values during hospitalization], and the level of respiratory support were also provided.

At admission, either X-rays or thoracic computed tomography (CT) scans were obtained for each patient. The diagnoses of PNM were confirmed by chest radiography or CT scans. An analysis of chest CT imaging included an assessment of the following lesion features: extent, distribution, the number of involved lobes, disease patterns (e.g., ground-glass opacities [GGOs], consolidations, crazy paving pattern), and other findings (e.g., a pneumothorax, pleural effusion, and pulmonary emphysema).

Concerning outcomes, we prospectively collected data on the length of hospital stay, number of days from diagnosis to development of PNM, need for endotracheal intubation (ETI), and in-hospital death.

We used descriptive statistics to report the demographic and clinical characteristics of different groups. Differences between groups were tested using Fisher’s exact test for categorical variables and Student’s *t*-test (the Mann–Whittney test), as appropriate, with an α = 0.05 significance level. Since the control group was not matched for gender and age, these parameters were included in the analyses. All analyses were performed using GraphPad Prism 9.0.0.

## 3. Results

We recruited a total of 65 severe COVID-19 patients, among whom 32 (49%) had PNM and 33 (51%) did not develop PNM (controls).

Demographic and clinical characteristics of patients of the study sample (n = 65) are described in Table 1.

The mean age was significantly lower in the PNM group (54.9 ± 18.5) compared to the No PNM group (65.4 ± 14.3, *p* = 0.0214), suggesting that younger patients might be more susceptible to developing PNM during severe COVID-19 disease. However, gender distribution did not show a significant difference (*p* = 0.5977), indicating that both men and women are equally likely to develop PNM when infected with SARS-CoV-2.

Smoking status did not significantly differ between the groups in our study. Current smokers were slightly more prevalent in the PNM group (40.6% vs. 24.3%, *p* = 0.1912), but this difference was not statistically significant. Similarly, BMI did not differ significantly between the average of the two groups (No PNM: 27.7 ± 6.5; PNM: 26.5 ± 4.9; *p* = 0.5727), with overweight and obesity being common in both groups. A higher BMI tended to show a protective effect against PNM due to COVID-19 (*p* = 0.0131). Only two patients were underweight, all in the case group.

Significant differences in inflammatory markers and WBCs were observed. The PNM group demonstrated significantly higher WBCs both at presentation (11.4 vs. 6.5, *p* < 0.0001) and during their hospital stay (16.4 ± 6.1 vs. 8.9 ± 5.1, *p* < 0.0001), suggesting a heightened inflammatory response. This elevation may be attributed, in part, to the development of nosocomial infections with Acinetobacter or Klebsiella pneumoniae in 10 patients (31.5%) from the PNM group during hospitalization. However, maximum CRP levels (No PNM: 126.3 ± 123.6 mg/L; PNM: 139.4 ± 85.9 mg/L; *p* = 0.3584) and D-dimer levels (7467 ± 12,243 ng/mL vs. 5497 ± 10,060 ng/mL; *p* = 0.7195) at presentation did not differ significantly between the groups, indicating that these markers alone may not be predictive of PNM development in patients with severe COVID-19 disease.

Our study did not find significant differences in the prevalence of hypertension (55.4%), coronary artery disease (20%), diabetes (15.4%), and obesity (21.5%) between the PNM and No PNM groups. Among the pre-existing respiratory conditions, COPD was present in 20 (30.7%) patients, bronchiectasis in 5 (7.7%) and pulmonary fibrosis in 5 (7.7%). Interestingly, chronic respiratory conditions such as COPD showed a significant prevalence difference between the two groups (No PNM 15.1% vs. PNM 46.9%, *p* = 0.0148).

Radiological findings were notable in both groups (see Table 2), with a significant portion of patients showing extensive lung involvement. Ground-glass opacities (GGOs) were the most frequent parenchymal alterations, present in 72.3% of patients, with a higher prevalence in the PNM group (93.7% vs. 51.5%). Additionally, consolidations (78.1%) and pleural effusions (28.1%) were more prevalent in the PNM group, suggesting that severe lung involvement could be a risk factor for developing PNM. In 52.3%, a chest CT scan revealed the involvement of more than three lobes in both groups. Also, 62.5% of patients in the PNM group developed a pneumothorax within 12.3 ± 14.9 days from the diagnosis of SARS-CoV-2 that required chest tube positioning.

The longer hospital stay observed in the PNM group (28.5 vs. 12.0 days, *p* < 0.0001) indicates a more severe clinical course. Despite this, the mortality rate did not significantly differ between the groups [45.5% in the No PNM and 42.4% in the PNM (*p* = 0.8901)]. In addition, the high mortality rates reflect both the high mortality impact of the “first wave” of COVID-19 infection in Italy and probably the higher clinical criticality of these patients regarding the COVID-19 population.

The mean interval from SARS-CoV-2 diagnosis to PNM occurrence was 11.6 ± 13.6 days, with a substantial proportion of PNM cases (24.2%) present at admission and 66.7% occurring within 5 days. The relatively rapid resolution of PNM (a mean time of 9.2 ± 4.6 days) in 16 (50%) patients highlights the potential for recovery. However, the significant association of PNM with the need for invasive mechanical ventilation (IMV) in 17 patients (53.1%) and the prevalent use of non-invasive respiratory support (NIRS) in 23 patients (71.8%) underscores the critical need for vigilant respiratory support in these patients. Among those requiring IMV, most were managed with volume-controlled ventilation (VCV) or pressure-controlled ventilation (PCV), with the approach tailored to each patient’s clinical condition. However, other detailed data regarding IMV, such as specific settings or adjustments, were not available or had limited retrieval possibility. The median duration of mechanical ventilation for both groups was 12 days (IQR 8–18 in the PNM), while in the No PNM group, it was 8 days (IQR 5–12). This extended duration in the PNM group likely reflected more severe clinical courses and complications that required prolonged respiratory support. All data for respiratory support are shown in Table 3.

## 4. Discussion

The emergence of PNM as a complication in patients with severe COVID-19 disease presents significant challenges and implications for clinical management. This study’s findings, highlighting risk factors and clinical outcomes associated with PNM in critical COVID-19 patients, offer a deeper understanding of the condition’s pathophysiology and potential strategies for mitigation.

Generally, in COVID-19 patients, severe inflammation and lung tissue damage due to viral infection, coupled with the increased intrathoracic pressures from vigorous coughing or mechanical ventilation, may predispose patients to alveolar rupture [4]. This phenomenon is probably even more evident in those patients with severe COVID-19 disease. Indeed, the presence of extensive lung involvement (≥50%) and consolidations observed in this study further supports the idea that severe lung pathology is a critical factor in the development of PNM.

Our study demonstrates that younger age may be a significant risk factor for the development of PNM in severe COVID-19 patients, with the PNM group having a mean age approximately 10 years younger than the control group, suggesting that age alone may not be a protective factor against the development of PNM. This aligns with previous studies [5] suggesting that younger patients may be more prone to severe lung injury and air leakage syndromes, possibly due to a more vigorous immune response or higher levels of physical activity, that may contribute to higher intrathoracic pressures from vigorous coughing, predisposing them to alveolar rupture. Rafiee et al. reported that five patients with COVID-19 pneumonia developed SPT and one developed SPM, and male gender was one of the risk factors for these complications [6]. In our study, gender did not emerge as a significant risk factor; however, in both groups, more than three out of five patients were male.

Contrary to some prior reports [3], our study did not find a significant difference in smoking status between the PNM and No PNM groups, although there was a non-significant trend toward higher smoking rates in the PNM group. This finding suggests that smoking increases severity and mortality of both bacterial and viral infections through the induction of mechanical and structural changes in the respiratory tract and alteration of cell- and humoral-mediated immune responses [7]. This may indicate that smoking is not a primary driver of PNM in COVID-19, and other factors, such as the severity of lung involvement or the mechanical forces of respiration, could play a more critical role. Thus, smoking cessation and the close monitoring of patients with a significant smoking history are essential components of managing COVID-19 patients at risk of PNM.

Interestingly, our analysis identified a potential protective effect of higher BMI against the development of PNM, with a lower prevalence of obesity in the PNM group and a significant association between lower BMI and PNM development. This finding contrasts with the common understanding of obesity as a risk factor for severe COVID-19 outcomes, suggesting that the relationship between body composition and PNM may be more complex [3]. The presence of underweight individuals exclusively in the PNM group further supports this hypothesis. Malnutrition, including deficiencies related to low BMI, can impair immune function and respiratory mechanics, increasing the risk of complications in critically ill COVID-19 patients. Deficiencies in nutrition associated with low BMI might lead to a deficiency in α1-antitrypsin, and α1-antitrypsin deficiency could promote damage of the bronchial wall [8,9]. The hypotheses to explain this trend include an immunologic protective effect of estrogen in females, a greater smoking prevalence among males, and the androgen-driven pathogenesis of COVID-19 [10].

The significantly higher WBCs observed in the PNM group suggest that an elevated inflammatory response may contribute to the pathogenesis of PNM in severe COVID-19 disease, where severe inflammation and lung tissue damage due to viral infection are prevalent. It has been demonstrated that in severe COVID-19 cases, the hyper-inflammatory process may result in alveolar damage and rupture [11]. The complications in our patients may be attributed to this mechanism. Pavlou et al. (2021) discuss how elevated inflammatory markers can predict complications in COVID-19, underscoring the importance of monitoring biomarkers to identify patients at risk of severe outcomes such as pneumomediastinum [12].

Lymphopenia, ferritin, CRP, lactate dehydrogenase (LDH), and D-dimer elevation are widely recognized as poor prognostic factors in the general population of COVID-19 patients [13]. These biomarkers may lead to multi-organ damage or ARDS, resulting in a cytokine storm. Also, a higher Sequential Organ Failure Assessment (SOFA) score and elevated d-dimer levels at admission were associated with higher mortality in patients with COVID-19, emphasizing the importance of the early identification of at-risk individuals for targeted interventions [14,15]. However, in our study, we observed a different pattern. There was no significant difference in maximum C-reactive protein (CRP) levels and D-dimer concentrations, both at admission and during hospitalization, between the groups, indicating that these markers alone may not be sufficient to predict the development of PNM, underscoring the multifactorial nature of its pathogenesis. This highlights the need for further investigation into other inflammatory pathways and markers that could better differentiate patients at risk of this complication.

Among the pre-existing conditions examined, chronic obstructive pulmonary disease (COPD) was significantly more prevalent in the PNM group, with nearly half of the patients affected. This finding is consistent with the notion that an underlying lung disease predisposes patients to complications such as PNM, particularly in the context of severe viral pneumonia like COVID-19 [1]. A review by Reddy, S. Y. and Shah, P. (2021) highlights that while COPD is a notable risk factor, other chronic conditions, such as hypertension and diabetes, also influence COVID-19 severity, though their impact is less pronounced compared to that of chronic respiratory diseases [16]. However, other common comorbidities, including hypertension, diabetes, and coronary artery disease, did not show significant differences between the groups, suggesting that these conditions do not specifically predispose patients to PNM.

Radiological findings provided crucial insights into the relationship between lung involvement and PNM. Moghaddam and Crouser (2020) provide a comprehensive analysis of the role of CT imaging in the management of COVID-19, emphasizing its critical role in predicting and assessing complications associated with the disease. Their study underscores that CT imaging is invaluable for evaluating the extent of lung damage and guiding therapeutic decisions in COVID-19 patients. Specifically, they highlight that detailed CT findings, such as ground-glass opacities and consolidations, can reveal the severity of lung involvement, which is essential for predicting potential complications such as PNM [17]. In fact, ground-glass opacities (GGOs), consolidations, and pleural effusions were significantly more common in the PNM group, indicating that severe lung damage may be a key driver of PNM development in COVID-19 patients, requiring an aggressive and careful respiratory management. Similar results also emerged in a study by Khaire et al., where PNM or pneumothorax (PNX) incidence was higher in patients with a worse 25-point CT severity score [18,19]. Combining the assessment of imaging features with clinical and laboratory findings could facilitate the early diagnosis of COVID-19 pneumonia [20,21]. The high incidence of a pneumothorax and emphysema in the PNM group further emphasizes the vulnerability of these patients to air leakage syndromes, likely due to extensive alveolar damage and air trapping. The extent of lung involvement, although not significantly different between the groups, was still substantial, with about three lobes involved in the majority of cases. This widespread lung damage likely creates a predisposition to air leaks, leading to PNM and other complications such as a pneumothorax, which was observed in over 60% of the PNM group. These findings support the notion that close monitoring and timely intervention are essential for managing respiratory complications in COVID-19 patients, particularly those with extensive lung involvement [22].

It has been reported that the onset time of COVID-19-related PNX and PNM and the hospitalization of patients was on average 9–19.6 days and invasive mechanical ventilation was 5.4 days later. In seven patients, radiological evidence of spontaneous air leakage was found with a mean of 9 ± 9 days after the onset of COVID-19 symptoms and a mean of 6 ± 8 days after hospitalization [23]. Similarly in our study, the mean interval from SARS-CoV-2 diagnosis to PNM occurrence was 11.6 ± 13.6 days, with a substantial proportion of PNM cases (24.2%) present at admission and 66.7% occurring within 5 days. This aligns with the study conducted by Wang, M., Liu, Y. and Zhang, X. (2022) [24]. The longer hospital stays and high intubation rates observed in individuals in the PNM group reflect the increased severity and complexity of their clinical course and outcomes. Despite this, the mortality rate did not differ significantly between the groups (despite being very high), which could be attributed to effective management strategies, including the prompt identification and treatment of PNM. The substantial proportion of PNM cases occurring within days of a COVID-19 diagnosis suggests that early and vigilant monitoring is crucial, particularly in patients requiring high levels of respiratory support such as invasive mechanical ventilation (IMV) or high-flow nasal cannula (HFNC) [25]. The early use of non-invasive ventilation (NIV), particularly continuous positive airway pressure (CPAP) and bi-level positive airways pressure (BiPAP), has been associated with an increased risk of barotrauma in patients with compromised lung function, such as those with severe COVID-19 pneumonia. Barotrauma occurs when excessive positive pressure causes alveolar rupture, leading to conditions like PNM or PNX [2]. However, in our cohort, we observed that while NIV was used as the initial intervention in some patients with PNM, its role in the immediate development of PNM was limited. Most cases of PNM in our study appeared later in the disease course, often after an escalation to IMV. These findings align with previous studies suggesting that IMV, especially with higher pressures, is a more common cause of barotrauma in critically ill COVID-19 patients [26]. The temporal relationship in our cohort suggests that factors such as severe lung involvement, the progression of respiratory failure, and the need for IMV played a more prominent role in the development of PNM rather than NIV alone [25]. While NIV can contribute to barotrauma, its application in the early stages did not consistently correlate with the immediate onset of PNM, highlighting the complex and multifactorial nature of PNM pathogenesis in severe COVID-19.

The relatively rapid resolution of PNM in half of the affected patients is a positive finding, indicating that with appropriate care, recovery is possible. However, the significant association of PNM with the need for IMV underscores the critical nature of this complication and the importance of tailored respiratory support to mitigate its impact in order to minimize barotrauma.

The pathogenesis of PNM in severe COVID-19 disease is likely multifactorial, involving severe lung inflammation and damage, increased intrathoracic pressures from coughing or mechanical ventilation, and pre-existing pulmonary vulnerabilities [27,28]. The identification of risk factors such as extensive lung involvement, a low BMI, high inflammatory markers, and smoking exposure can aid clinicians in the early detection and intervention of the disease, potentially mitigating adverse outcomes [29,30].

In conclusion, PNM represents a significant and major complication in patients with severe COVID-19 disease, associated with worse clinical outcomes and increased morbidity. Through ongoing research and adaptive clinical strategies, healthcare providers can better navigate the complexities of COVID-19 and enhance care for patients affected by PNM.

Further research is essential to refine our understanding of PNM in COVID-19 patients. Larger studies can validate the identified risk factors and explore additional variables that may influence PNM development and outcomes. Investigating the effectiveness of different management strategies, including the role of various respiratory support modalities and anti-inflammatory treatments, will be crucial in developing evidence-based guidelines.

Moreover, as the pandemic continues to evolve, there is a need to adapt and refine clinical protocols to address emerging complications like PNM. The continuous monitoring of patient outcomes and incorporating new research findings into clinical practice will be key to improving the prognosis for COVID-19 patients experiencing this severe complication.

This study had several limitations. First, despite the high number of overall COVID-19 patients the number of cases analyzed was relatively small to perform robust statistical evidence. Second, the study was a retrospective, single-center study conducted in a period where the health system in Italy was in shock. Thus, several biases and missing data may be justified. Finally, the selection of patients (“severe COVID-19 disease”) was based only on “frail” clinical reports (respiratory failure) and on the hospitalization in critical care units with no rigid parameters. In addition, the lack of clear risk factors with statistical significance could be attributed to the relatively small sample size of our study. Thus, further studies with large sample sizes are needed to clarify the risk factors associated with PNX/PNM in patients with severe COVID-19 disease. Also, it is difficult to predict whether our results will apply to current and future landscapes, characterized by new variants of SARS-CoV-2 and the immunization of many people due to vaccination and previous exposure.

## 5. Conclusions

Pneumomediastinum is an increasingly recognized but severe complication in critical COVID-19 patients, highlighting the complex interplay of risk factors that predispose certain individuals to this condition. This study elucidates the significant clinical impact of PNM as a complication in patients with severe COVID-19 disease, underscoring the need for heightened awareness and targeted management strategies. The findings demonstrate that PNM is associated with younger age, higher inflammatory markers, and extensive lung involvement, highlighting the multifactorial pathogenesis involving severe lung inflammation, increased intrathoracic pressures, and pre-existing pulmonary vulnerabilities.

In light of these findings, it is crucial to continue exploring the pathophysiology of PNM in COVID-19 patients, particularly as the pandemic evolves and new variants of SARS-CoV-2 emerge. Further research with larger cohorts is essential to validate the risk factors identified in this study and to develop evidence-based management strategies that can improve outcomes for patients at risk of PNM.

## Figures and Tables

**Figure 1 diseases-12-00242-f001:**
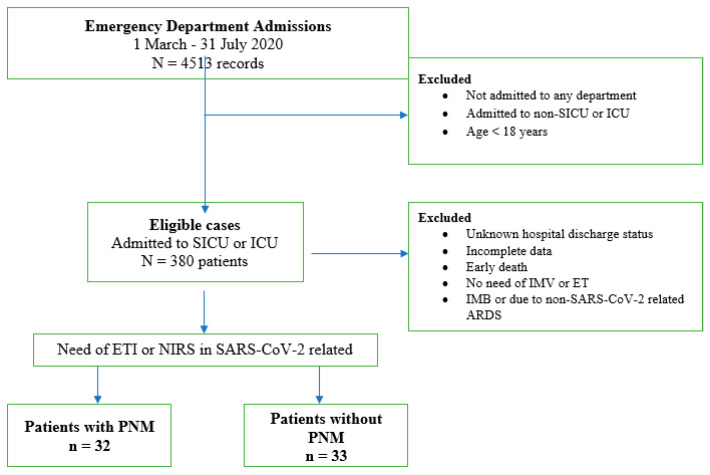
Study population selection. SICU: sub intensive care unit; ICU: intensive care unit; IMV: invasive mechanic ventilation; ETI: endotracheal intubation; NIRS: non-invasive respiratory support; ARDS: acute respiratory distress syndrome; PNM: pneumomediastinum.

**Table 1 diseases-12-00242-t001:** Demographic and clinical characteristics of patients with severe COVID-19 disease (n = 65).

	No PNM (n = 33)	PNM (n = 32)	*p*-Value
Men	21 (63%)	23 (72%)	0.5977
Age (years) *	65.4 ± 14.3	54.9 ± 18.5	0.0214
Smoking exposure			
Never a smoker	11 (33.3%)	9 (28.1%)	0.7893
Former smoker	14 (42.4%)	10 (31.3%)	0.4431
Current smoker	8 (24.3%)	13 (40.6%)	0.1912
Nutritional status			
BMI *	27.7 ± 6.5	26.5 ± 4.9	0.5727
Underweight (BMI < 18.5)	0	2 (6.3%)	0.2385
Overweight (BMI 25–29.9)	17 (51.5%)	18 (56.3%)	0.8050
Obese (BMI > 30)	10 (30.3%)	4 (12.5%)	0.0131
Blood parameters			
CRP at admission (mg/L) *	73.6 ± 108.7	108.1 ± 87.6	0.0187
Maximum CRP (mg/L) *	126.3 ± 123.6	139.4 ± 85.9	0.3584
D-dimer at admission (ng/mL) *	7467 ± 12,243	5497 ± 10,060	0.7195
Maximum D-dimer (ng/mL) *	10,412 ± 13,075	8354 ± 10,150	0.9958
WBC at admission	6.5 ± 4.1	11.4 ± 5.4	<0.0001
Maximum WBC	8.9 ± 5.1	16.4 ± 6.1	<0.0001
Comorbidities			
Hypertension	20 (60.6%)	16 (50%)	0.4586
COPD	5 (15.1%)	15 (46.9%)	0.0148
Diabetes mellitus type II	7 (21.2%)	3 (9.4%)	0.3020
CAD	7 (21.2%)	6 (18.6%)	>0.0999
Other respiratory diseases (fibrosis and bronchiectasis)	7 (21.2%)	5 (15.6%)	0.7505

* Values are expressed as mean ± standard deviation. BMI: body mass index; CRP: C-reactive protein; WBC: white blood cell; CAD: coronary artery disease; COPD: chronic obstructive pulmonary disease.

**Table 2 diseases-12-00242-t002:** Chest CT scans at admission in the two groups.

	No PNM (n = 33)	PNM (n = 32)	*p*-Value
Extent of lung involvement *	52.7 ± 27.4 (1–100)	61.3 ± 20 (1–100)	0.3278
Number of lobes involved *	2.6 ± 1.1 (1–5)	2.6 ± 0.9 (1–5)	0.8294
Bilateral distribution	29 (87.9%)	28 (87.5%)	>0.9999
Ground-glass opacities	17 (51.5%)	30 (93.7%)	0.0002
Consolidations	14 (42.2%)	25 (78.1%)	0.0051
Crazy paving	6 (18.2%)	3 (9.4%)	0.4752
Pleural effusion	3 (9%)	9 (28.1%)	0.0606
Pneumothorax	1 (3%)	20 (62.5%)	<0.0001
Emphysema	0	6 (18.6%)	0.0110

* Values expressed as mean ± standard deviation (range).

**Table 3 diseases-12-00242-t003:** Respiratory support in the two groups.

	No PNM (n = 33)	PNM (n = 32)	*p*-Value
Conventional OT	13 (39.4%)	9 (28.1%)	0.4339
CPAP	4 (12.1%)	8 (25%)	0.3389
BiPAP	10 (30.3%)	8 (25%)	0.1548
HFNC	6 (18.2%)	7 (21.9%)	0.7642
ETI	10 (30.3%)	17 (53.1%)	0.0619

OT: oxygen therapy; CPAP = continuous positive airway pressure; BiPAP: bi-level positive airway pressure; HFNC: high-flow nasal cannula; ETI: endotracheal intubation.

## Data Availability

Data are available from the authors upon request.

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
