# Peer review of "Unraveling Pneumomediastinum in COVID-19 Patients: Insights from a High-Volume-Center Case–Control Study"

_diseases, 2024, doi:10.3390/diseases12100242_

Round 1

Reviewer 1 Report

Comments and Suggestions for Authors

Brief summary

The study uses retrospective clinical data from patients with severe COVID-19 with and without PNM to identify risk factors, clinical characteristics, and outcomes in this patient population. The authors found that younger age, elevated inflammatory markers, and extensive lung involvement were associated with PNM in patients with severe COVID-19.

General concept comments

The introduction and discussion are nicely written. However, within the methods and results section several sentences and paragraphs display average to poor english writing. I would recommend to have these chapters reviewed by a native speaker for fluent and precise writing which in turn leads to easier reading and understanding. 

Within the text neither Figures nor Tables are adequately referenced (sometimes only at the end of a paragraph). Tables and Figures should be referenced whenever values in the text correspond to a table/figure. Please revise. 

Specific comments

Line 24: should also be retrospective regarding the data collection (not prospective), since data was already available and used for this analysis. 

Line 63: similar (not similarly)

Line 68/69: The second sentence does not seem to be complete; maybe "In Figure 1 the consort diagrams displays the patient population analysed for this study" (or something like that.

Figure 1: In the first "excluded" box: I assume that is supposed to be <18 years? Otherwise this diagram would state that you only included patients younger than 18 years. In the second "excluded" box: Last bullet points: Should that be "IMV" - otherwise explanation for abbreviation IMB is missing.  An is that supposed to mean "IMV due to non- SARS-CoV-2 related AEDS"? If so, the "or" should be removed.

Line 76-78: the number 4513 is mentioned twice in this sentence - shouldn`t the second number be 380. Please revise the sentence. In detail (without the s). 

Line 78/79: were admitted to the emergency department...

Line 82: Which diagnosis? 

Table 1: Blood parameters: either write "at admission" or "at presentation" for all blood parameters documented at admission to be consistent. 

Line 123: the two groups (not thew two groups)

Line 134: That is not entirely true. There is a significant difference in the prevalence of COPD.

Line 136: I think there is something missing after among. Among whom?

Line 137: 20 patients (not 20 of patients)

Line 138/139: This sentence contradicts the first sentence on the paragraph. Please revise paragraph/sentence accordingly. 

Line 148: alterations

Line 152: of patients in the PNM group

Table 3: Why is IMV not displayed in the table as well? Should be added.

Table 3 and Line 166: Values for HFNC in the text and the table are different - 41% in the text and 21.9% in the table. Please revise. 

Line 223: It has been demonstrated

Line 228: no significant difference was observed (not wasn`t a)

Line 302: The second " is missing. 

Comments on the Quality of English Language

As already pointed out above, I would recommend to have especially the methods and results section reviewed by a native speaker to improve the writing.

Author Response

Thank you for this insightful comments. You can find all the answers and corrections in the attached file belowe. 

Reviewer 2 Report

Comments and Suggestions for Authors

Although not innovative because update on pneumomediastinum appearead early during first phase of pandemic, authors report an intriguing cohort of patient.

from a clinical poin of view I raise only one question about risk for pneumomediastinum : they did not report the early use of NIV as cause of pneumomediastinum per se.

they should add this item to text and tables and discuss it.

Author Response

Dear reviewer, thanks for the time You spend for performing these constructive revisions. In the attached file, you may find our point-by-point replies.

Thanks in advance for Your kind support

Reviewer 3 Report

Comments and Suggestions for Authors

The authors assessed the impact of a rather rare complication of pneumomediastinum (PNM) on the course of COVID-19 infection. Two groups of patients with and without PNM were compared. Based on the data  it was concluded that although there were no differences in mortality, patients with PNM required more intensive treatment.

Major comments:

1. The manuscript does not describe in sufficient detail the severity of the condition of patients in each of the groups. Since PNM is most often associated with barotrauma, it is necessary to indicate what type of invasive mechanic ventilation was used in each of the groups and what is the median duration of ventilation.

2. According to the presented data on inflammatory markers, WBC and the number of pneumothoraxes in WBC patients, a more severe course of the disease was initially noted.

Minor comments

1. It is advisable to use non-parametric criteria for comparing parameters with the definition of the median, 25 and 75 quadrilaterals. The average values ​​may be inaccurate due to the lack of normal distribution of the data. These doubts are based on the lack of differences in the maximum values ​​of some parameters.

2. The author demonstrates a high level of leukocytosis in the group with PNM (WBC max. 16.4) and does not provide data on the associated bacterial infection.

3. It is not clear what arithmetic mean error means (for example, WBC max. 16.4±6.1, if the maximum value of the parameter is given?

4. The authors mention the prognostic significance of lymphopenia in COVID-19, but provide this data.4. The authors mention the prognostic significance of lymphopenia in COVID-19, but provide this data.

Author Response

(The authors gave the same response as above.)

Reviewer 4 Report

Comments and Suggestions for Authors

The paper reported a single-center, retrospective, case-control study. The authors analyzed 4513 patients in their hospital and focused on 65 patients who underwent to Emergency department with severe disease (respiratory failure) and were hospitalized in critical care units. They compared 33 patients who developed pneumomediastinum (PNM) and 32 who did not develop PNM.

However, in Table 1-3, No PNM is n = 33; PNM is n = 32, please check the results presented in the tables. The numbers in the tables are different from Figure 1 and the other Sections in the paper: In Section 2. Materials and Methods: Line 80: “The cohort included 33 patients who developed PNM during the hospital stay and 32 who did not.” In the Results section: Line 104-105: “We recruited a total of 65 severe COVID-19 patients, among whom 33 (51%) had PNM and 32 (49%) did not develop PNM (controls).

1. Please indicate the significance of the results in the tables.

2. Please Keep the reference format consistent.

Comments on the Quality of English Language

1.       Please provide full name for: Line 29 WBC; Line 30 BMI; COPD; Line 226: LDH; Line 246: PNX

2.       Please check the use of the abbreviations. For example, pneumomediastinum is abbreviated as PNM in Line 49, then PNM should be used for further uses, Line 256 pneumomediastinum should be PNM.

3. Please check through the text, keep a space before and after a symbol/unit.

Author Response

Herein the attached file wityh our revisions

Round 2

Reviewer 4 Report

Comments and Suggestions for Authors

Some minor comments: 

1. Please keep a space before and after a symbol:

In the tables:

Table 1, Table 2 and 3: n=33 → n = 33;  n=32 → n = 32

Table 2: 52.7±27.4, 61.3±20, 2.6±1.1, 2.6±0.9 → 52.7 ± 27.4, 61.3 ± 20, 2.6 ± 1.1, 2.6 ± 0.9

Please check through the paper for:

p=XX → p = XX

p<XX→ p < XX

2. Abbreviation use:

Line 187 pneumomediastinum (PNM) → PNM

Line 233 white blood cell (WBC) → WBC

 3. References:

Please provide the full journal name for Reference 3, 10 and 15.

Comments on the Quality of English Language

Please refer to above section.

Author Response

Revisions have benne summarized in the attached file
